# Multimodal Device for Real-Time Monitoring of Skin Oxygen Saturation and Microcirculation Function

**DOI:** 10.3390/bios9030097

**Published:** 2019-08-02

**Authors:** Uldis Rubins, Zbignevs Marcinkevics, Janis Cimurs, Inga Saknite, Edgars Kviesis-Kipge, Andris Grabovskis

**Affiliations:** 1Biophotonics Laboratory, Institute of Atomic Physics and Spectroscopy, University of Latvia, Raina Blvd. 19, LV-1050 Riga, Latvia; 2Department of Human and Animal Physiology, University of Latvia, Raina Blvd. 19, LV-1050 Riga, Latvia

**Keywords:** hyperspectral imaging, multimodal imaging, skin oxygen saturation, skin mottling, microcirculation

## Abstract

The present study introduces a recently developed compact hybrid device for real-time monitoring of skin oxygen saturation and temperature distribution. The prototype involves a snapshot hyperspectral camera, multi-wavelength illuminator, thermal camera, and built-in computer with custom-developed software. To validate this device in-vivo we performed upper arm vascular occlusion on eight healthy volunteers. Palm skin oxygen saturation maps were analyzed in real-time using k-means segmentation algorithm and two-layer optical diffuse model. The prototype system demonstrated a satisfying performance of skin hyperspectral measurements in the spectral range of 507–625 nm. The results confirmed the reliability of the proposed system for in-vivo assessment of skin hemoglobin saturation with oxygen and microcirculation.

## 1. Introduction

Over the last decades, there has been growing evidence that human cutaneous microcirculation can reflect different aspects of neuro-immuno-endocrine functions and may serve as an early marker for various pathological conditions such as diabetes [1,2], systemic rheumatic disease [3,4], neuropathy [5], and sepsis [6,7]. The disturbance of blood supply leads to impairments of cutaneous oxygen saturation, which is one of the most essential vital parameter in the assessment of tissue health. As a consequence blood pooling occurs, producing skin mottling, which is characterized by purple or reddish patches that cover the skin of legs, arms, or upper body. The classification and quantification of mottling are currently entirely subjective, based on visual inspection by the clinical expert [8]. At present, there are no appropriate objective mottling evaluation methods.

A potential technique for non-invasive assessment of mottled skin patches is hyperspectral imaging (HSI) [9]. The technique is based on the acquisition of multiple images within relatively narrow non-overlapping spectral bands. The light delivered to biological tissue undergoes multiple scattering and absorbance from inhomogeneities of biological structures. In the visible wavelength range, the main tissue absorbers are melanin, oxy- and deoxy-hemoglobin. Assessing the concentrations of these chromophores can be used to identify pathological changes in the skin. The advantage of HSI over the convenient diffuse reflectance spectroscopy (DRS) is the possibility of measuring the spatial distribution of skin chromophores as well as obtaining spectral information. Therefore, it can be applicable to skin mottling analysis.

Since blood circulation yields heat transfer, cutaneous microcirculation patterns can be monitored by thermography (TG) [10]. Typically, skin mottling is present in case of sepsis, where thermal images of patients with septic shock reveals heterogeneous patterns containing “hot-spots” on most of the locations (i.e., the arterial perforators) [11,12,13]. Such heterogeneous patterns reflect diminished microcirculatory perfusion in the superficial dermal plexus and produce temperature gradient between relatively cold skin and hot perforating arteries, which is a sign of impaired microcirculation and may serve as an early indicator of improvement or deterioration of the patient’s condition [11,12,13].

An optimal device for monitoring the microcirculatory function impairments should be non-invasive, portable, and suitable for patient’s bedside usage. Despite the HSI ability to obtain clinically relevant information, only a few commercial systems exist which can assess oxygen saturation [14,15]. These systems are costly and impractical for clinical use [15]. Due to the progress of opto-electronics, it is now possible to build a portable and easy-to-use imaging system for optical assessment of skin microcirculation, which can be used for medical applications. Several groups and companies have attempted to develop portable multispectral imaging systems, which can quantify scattering and absorption in the tissue [16,17,18,19,20,21,22,23,24]. However, to the best of our knowledge, there is no portable, hand-held, and robust hyperspectral device capable of assessing skin mottling and hemoglobin saturation with oxygen.

In the present work, we present a compact multimodal imaging device pre-commercial prototype (technology demonstrator) with a dual modality camera system. The prototype device involves a single-snapshot mosaic type camera for fast acquisition of HSI data and a thermal camera for skin temperature measurements. A built-in powerful mini-computer operated by custom developed software with integrated diffuse light-skin transport mathematical model can estimate skin mottling parameters in real time. To validate our system, we recorded in-vivo data using vascular occlusion physiological provocation tests on healthy subjects. During this provocation, physiological skin mottling was evoked, and consequently, skin oxygen saturation and temperature measurements were performed.

## 2. Materials and Methods

A multimodal prototype device was developed for real-time inspection of skin blood hemoglobin saturation with oxygen (*SO*_2_) maps and thermal images of skin. The device can perform calculations in real-time, and it is suitable for clinical use for bedside monitoring of septic patients. Here we described the design of the device, the image processing algorithm steps, and the software. We tested the proposed device on healthy volunteers using a vascular occlusion test applied to the upper arm.

### 2.1. The Prototype Device

The prototype device (Figure 1) comprised of three main modules: (1) A hyperspectral imaging (HSI) system, (2) a thermography (TG) system, and (3) a computing (CPU) and display module. All components were installed into a custom printed tube and rotational case. The HSI and TG systems were built in the tube housing in opposite direction. The CPU module with power supply and touch-screen display was built in a rectangular case at the side of tube between the HSI and TG systems, where the CPU module can be rotated over the horizontal axis for user convenience (Figure 1).

The HSI system was previously designed and tested in our laboratory [25]. Later it was re-designed for clinical needs. The HSI system can be subdivided into an illumination and detection module. The illumination module involves a set of visible-spectrum LEDs (Luxeon Rebel from Lumileds, San Jose, CA, USA, consisting of 6 cyan LEDs CW = 505 nm/FWHM = 30 nm, and 12 lime LEDs CW = 568 nm/FWHM = 100 nm), LED driver and 12-bit DAC controller. The DAC controller allows the control of the illumination intensity of all LEDs through the USB interface. All illuminator components were mounted on a printed circuit board. To provide uniform illumination of skin, a light diffuser (Edmund Optics) was placed in front of the illuminator ring. The linear polarizer film was installed in the front of the illuminator. Measured irradiance in the skin plane was 1 mW/cm^2^.

The detection module involves a compact snapshot HSI camera (XIMEA xiSpec MQ022HG-IM-SM4 × 4-VIS, 2048 × 1088 pix. snapshot, Fabry-Perrot interference filter 4 × 4 for 16 spectral bands in the range 470–630 nm, 10-bit ADC), which is equipped with near-field lens (Edmund Optics, Shenzhen, China, 8.5 mm HP Series Lens, 2/3” f/1.4) allowing skin measurements from the distance of 7 cm. Linear polarizer film was installed in the front of the lens. The camera polarizer was oriented orthogonally to the illuminator polarizer to eliminate skin glare. The illumination intensity of each spectral band was adjusted according to best-possible linearity of camera spectral sensitivity in the range of 500–625 nm. The field of view diameter of HSI system is 50 mm, spatial resolution is 0.18 mm. The TG camera (Xenics Gobi, Steemer Imaging, Puchheim, Germany, 640, 640 × 480 pix., 16-bit ADC, LWIR) was built on the other side of the tube. The field of view of camera is 150 × 112 mm, spatial resolution is 0.23 mm. The entire system can be controlled by built-in mini-computer (Windows-10 OS, CPU I-5, 16GB RAM, 512 GB SSD). A Graphical User Interface (GUI) developed in a Matlab environment allows control of the device through a 7” touch-screen display.

The Matlab GUI (Figure 2) allows the operator to preview skin images obtained from both cameras, perform real-time calculations, and visualize *SO*_2_ maps and skin temperature in color-coded image format, as well store data locally and remotely. When turned on, the multimodal device starts software automatically, which works in two camera modes: (1) HSI mode, and (2) TG mode. In the HSI mode, the illuminator switches on (intensity ratio between the cyan and lime LEDs set to 1/20), and HSI camera becomes active. By the default, the video is displayed in color-coded format (r-g-b composite video) in preview mode. In parametric mode, the saturation maps are displayed, showing diagnostic parameters. In the TG mode, the thermal camera becomes active, and the thermal video is displayed in the main window. Storing of data can be performed by the user during the monitoring. Data acquisition time in HSI mode is 80 ms (12.5 frames/s), processing time of saturation maps takes 1 s. The acquisition of TG data is performed by 25 frames/s

### 2.2. Measurement Protocol

The present study was approved by the Ethics committee of the University of Latvia and was conducted in accordance with the guidelines of the Declaration of Helsinki. Eight healthy subjects (25 ± 5 years old) were enrolled in the pilot study. Before the participation, the subjects were informed about the procedure and possible discomfort during the trial. Their written consent was obtained. During the recording procedure, the volunteers were in the sitting position, as the right palm was inspected. To eliminate movements during occlusion, the subject palm was fixed on a vacuum pillow support (AB, Germa, Sweden). For the evaluation of the prototype system, upper-arm suprasystolic vascular occlusion test was performed, including three stages: Baseline, arterial occlusion, and post-occlusive reactive hyperemia. In order to induce alterations of issue oxygen saturation, the upper arm was occluded (150 mmHg) with a cuff for 10 min. During the measurements, HSI data were recorded. The HSI side of the prototype tube is placed in contact with palm skin and data was acquired in HSI mode. During the thermal measurements, another end of the tube was oriented 7–12 cm apart from the skin surface, and the thermal image was recorded in TG mode.

### 2.3. The Processing of Data

The Matlab software performs the following operations: (1) The control of illuminator light intensity (default, in ratio 1/20 between the cyan and lime); (2) the control of HSI camera (adjustments of exposure, gain and frame rate), image acquisition; (3) TG camera control (temperature calibrations) and image acquisition; (4) real-time processing of HSI data; and (5) visualizations of diagnostic information (saturation maps and diagnostic parameters).

#### 2.3.1. HSI Data Preprocessing

In order to get HSI cube from RAW image data, every snapshot (2048 × 1088 pix.) was transformed into an image cube (width = 512, height = 272, bands = 16). The square part in the center (272 × 272 pix.) was extracted and the circular mask was applied to highlight the central area of interest of registered HSI data. Then, the following spectral channels were chosen from available 16 bands of HSI camera: CW = 507nm, 518 nm, 532 nm, 544 nm, 568 nm, 580 nm, 593 nm, 604 nm, 617 nm, 625 nm. The selection of spectral bands was based on minimal spectral mixing of camera bands and high absorption of hemoglobin. Spectral unmixing of image cube was done using linear transformation:(1)$$I=A\cdot I_{raw}^{-1}$$
where *I_raw_*(*x*, *y*, *λ*) is a raw pixel value obtained in every *xy* pixel at given wavelength *λ*, *I*(*x*, *y*, *λ*) is spectrally-unmixed data, *A*(*λ*) is tabulated Ximea sensor sensitivity data for 16 spectral channels (data rows) in wavelength interval 400–1000 nm (data columns). Sensor sensitivity data are provided by the Ximea manufacturer page. The matrix A is used to provide spectral unmixing of sensor raw data.

To assess the spectral response of the HSI sensor, we used a monochromator system. A broadband halogen light source (Thorlabs) was coupled into a monochromator input. The output of the monochromator was connected to HSI system input or spectrometer probe. The monochromator was scanned between 500 and 625 nm in 2 nm increments and intensity spectra was acquired via a calibrated inspection spectrometer (AvaSpec-ULS2048-USB2-FCDC, Avantes, Apeldoorn, The Netherlands). The FWHM of monochrome light was 4 nm. During the acquisition of HSI data, the measurement of each spectral band was saved as 10-bit tif image, which was analyzed by Matlab software in order to get Ximea sensor response curves.

In the development stage, the calibration to the reflectance *R*(*x*, *y*, *λ*) with dark *I_dark_* and white reference *I_white_*(*x*, *y*, *λ*) (99% spectralon reference standard X-Rite) was done:(2)$$R=0.99\cdot \frac{I-I_{dark}}{I_{white}-I_{dark}}$$

Equation (2) additionally removes uneven illumination.

#### 2.3.2. Feature Extraction from HSI Data

We chose k-means clustering algorithm [26] for HSI data classification due to its simple, fast, and fully-automatic capabilities. The main idea of the k-means algorithm is to divide *N* pixels of HSI (*N* = *W* × *H*) into a small number of *K* clusters. The *R*(*x*, *y*, *λ*) can be represented as a data cloud ***r*** in *B*-dimensional space (*λ* = 1, …, *B*). First, *K* centroids are defined randomly. Then the centroids *R_k_* can be iteratively found by minimizing the sum of Euclidean distance between the centroid *R_k_* and ***r*** for each cluster:(3)$$\underset{min}{arg}\overset{K}{\underset{k=1}{\sum }}\underset{r\in cluster_{k}}{\sum }\|r-R_{k}{\|}^{2}$$

The goal of the algorithm is finding of centroid location within each cluster, which represents the mean-averaged reflectance spectra *R_k_*(*λ*)(*k* = 1, 2, …, *K*), and obtaining the corresponding pixel locations in the HSI cube. In the model calculations (see results), we assumed optimal *K* = 10.

#### 2.3.3. Diffuse Model

In order to determine diagnostic parameters of the skin, a light-skin interaction model was elaborated. The analytical skin model is based on a simplified two-layered structure, the first one being the melanin containing epidermis, the second one being the blood containing dermis. The reflectance *R* depends on absorption and scattering properties of the different tissue layers in the skin [23]. The tissue absorption properties are specified as:*B*—the blood volume fraction (*B* = 1 for whole blood, 150 g hemoglobin/liter).*SO*_2_—the oxygen saturation of hemoglobin in blood.*M*—the volume fraction of typical cutaneous melanosomes in the epidermis.
which are used to calculate the absorption coefficient:(4)$$\mu_{a}\left(\lambda \right)=M\cdot \mu_{mel}\left(\lambda \right)+B\cdot SO_{2}\cdot \mu_{oxy}\left(\lambda \right)+B\cdot \left(1-SO_{2}\right)\cdot \mu_{deoxy}\left(\lambda \right)$$
where *µ_mel_* are tabulated from [27] and *µ_oxy_*, *µ_deoxy_* are found from [28].

The scattering properties of tissue are given by:*µ’_s_*_500 nm_—the reduced scattering at 500 nm.*f_Ray_*—the fraction of scattering at 500 nm due to structures much less than 500 nm (Rayleigh scattering).*(1-f_Ray_)*—the fraction of scattering at 500 nm due to structures comparable or larger than 500 nm (Mie scattering).*b_Mie_*—the scattering power for Mie scattering (*b_Mie_* < 1)
which are used to calculate the reduced scattering coefficient for any skin layer [29]:(5)$$\mu_{s}^{\prime }\left(\lambda \right)=\mu_{s500nm}^{\prime }\left(\lambda \right)\cdot \left[f_{Ray}{\left(\frac{\lambda }{500}\right)}^{-4}+\left(1-f_{Ray}\right){\left(\frac{\lambda }{500}\right)}^{-b_{Mie}}\right]$$

Our tissue model includes the scattering parameters above. In order to find model parameters, the inverse problem was solved, where a modeled diffuse reflection curve *R_model_* was fitted to the measurement data *R*. Using the diffusion model for two-layered skin described in [30], the modeled function can be described by the following equation:(6)$$R_{model}=A\delta_{e}\mu_{se}^{\prime }\frac{\Gamma_{1}+\Gamma_{2}}{\Gamma_{3}}$$
where:$$\Gamma_{1}=\left(1+\frac{\delta_{d}}{3\varsigma_{d}}\right)\left[\left(\frac{\delta_{e}^{2}\delta_{d}}{3}-\delta_{e}^{2}\varsigma_{d}\right)cosh\left(\frac{d_{e}}{\delta_{e}}\right)+\left(\frac{\delta_{e}^{3}\varsigma_{d}}{3\varsigma_{e}}-\delta_{e}\delta_{d}\varsigma_{e}\right)sinh\left(\frac{d_{e}}{\delta_{e}}\right)\right]$$
$$\Gamma_{2}=\left[\delta_{d}^{2}\varsigma_{e}\frac{\mu_{sd}^{\prime }}{\mu_{se}^{\prime }}\left(\frac{\delta_{e}^{2}}{9\varsigma_{e}^{2}}-1\right)+\delta_{e}^{2}\left(\varsigma_{d}-\frac{\delta_{d}^{2}}{9\varsigma_{d}}\right)\right]e^{-\frac{d_{e}}{3\varsigma_{e}}}$$
$$\Gamma_{3}=\left(\frac{\delta_{e}^{2}}{9\varsigma_{e}^{2}}-1\right)\left(\frac{\delta_{d}}{3\varsigma_{d}}+1\right)\left[\varsigma_{e}\delta_{e}\left(\varsigma_{d}+\delta_{d}A\right)cosh\left(\frac{d_{e}}{\delta_{e}}\right)+\left(\varsigma_{e}^{2}\delta_{d}+\varsigma_{d}\delta_{e}^{2}A\right)sinh\left(\frac{d_{e}}{\delta_{e}}\right)\right]$$
$$\varsigma =\frac{1}{3\left(\mu_{s}^{\prime }+\mu_{a}\right)}\to \;\delta =\frac{1}{\sqrt{\left(\mu_{s}^{\prime }+\mu_{a}\right)\mu_{a}}}$$

*d_e_*—is a thickness of epidermis layer.

The subscripts *e* and *d* under the symbols indicate the epidermis or dermis layer, respectively.

The model calculations were done in Matlab software using the non-linear optimization *lsqcurvefit* function. The parameters, which were used in model calculations, are shown in Table 1. The following model parameters were varied: *B*, *SO*_2_, *M*, and *d_e_*, while other parameters remain fixed.

#### 2.3.4. The Diagnostic Parameters

For each cluster, the *SO*_2_ were calculated, and the result was represented as a 2-*D* segmented oxygen saturation map. To characterize the morphology of skin color patches, two important parameters were found: (1) Subcutaneous tissue oxygen saturation *SO*_2_, and (2) the skin mottling parameter, which is related to *SO*_2_ changes over the skin. The main diagnostic parameter can be found as weighted mean *SO*_2_ value, which is defined as:(7)$$SO_{2\;mean}=\overset{K}{\underset{i=1}{\sum }}w_{i}\cdot SO_{2i}$$

The parameter *w_i_* is a weight factor for each cluster of saturation map, where *i* = 1:*K*, *K* is a number of clusters. The weight *w* depends on relative count of pixels included in corresponding cluster, and can be found by the equation: *w_i_* = *A_i_*/*A_RoI_*, where *A_i_* is an area of *i*-th cluster (count of pixels), *A_RoI_* is an area of all visible region of interest (count of pixels inside the disk area which is enclosed by rectangular RoI, see Figure 2b).

The skin mottling parameter can be found as a standard deviation of *SO*_2_:(8)$$SO_{2\;SD}=\sqrt{\frac{1}{K-1}\overset{K}{\underset{i=1}{\sum }}w_{i}\left(SO_{2i}-SO_{2\;mean}\right)}$$

## 3. Results

### 3.1. Sensor Calibration Data

Figure 3 shows the spectral responses of the HSI camera. The sensitivity of HSI camera was detected in the spectral range 500–625 nm, using monochrome illumination for the wavelengths which correspond to the 5th to 14th spectral channels of the HSI sensor. The quality of spectral response was measured by calculating area under curve (AUC) of normalized spectra. When compared the original responses to the spectrally unmixed responses (using sensor sensitivity data from Ximea datasheet and Formula (1)), the decrease of AUC by factor 1.5 ± 0.5 was observed in unmixed data, which confirmed the improvement of the quality of spectral response after the unmixing procedure.

### 3.2. Skin Oxygen Saturation

Figure 4 shows the set of images and saturation maps of palmar non-glabrous skin of 8 subjects. The in vivo assessment revealed the performance of the HSI system during real conditions using viable skin. During the baseline stage, all subjects exhibited healthy and slightly reddish color skin appearance, indicating typical perfusion and functional capillary density in this site. The oxygen saturation maps show the even distribution of *SO*_2_, which reached nearly 100%. During vascular occlusion, skin appearance underwent substantial changes—the occluded region of palm gradually changed color, becoming dark purple with white discoloration regions, due to the changes of proportion between oxygenated and deoxygenated blood and blood volume pooling within dermal cutaneous vessels. Such mottled skin was evoked in all subjects; however, intensity and patch size slightly varied among subjects. After releasing the cuff during the post-occlusion stage, extensive hyperemia was observed in the occluded region, as the blood flushed into the tissue, which became redder as the number of functional capillaries increased, leading to increase of total hemoglobin concentration.

The mean-averaged *SO*_2 *mean*_ values from 8 subjects during different vascular occlusion stages are summarized in Table 2. Acquired values demonstrate high similarity at the baseline and post-occlusive reactive hyperemia, when the tissue has a high concentration of oxygenated blood, while large differences among the subject group were observed in occlusion stage (at low concentrations of oxyhemoglobin). The skin mottling parameter *SO*_2 *SD*_ for the group showed the same tendency—high similarity in baseline and hyperemia and high deviation.

Our results imply that total hemoglobin concentration influences the HSI system accuracy—markedly higher errors at low concentrations of oxyhemoglobin (occlusion stage), suggesting that HSI analysis of pale skin is still challenging.

### 3.3. Skin Temperature Distribution

During the hyperemia stage, thermal images displayed specific “hot-spots”—areas with higher temperature, which gradually spread until they stretch to the whole palm. At the same time, the thermal gradient (heterogeneity) diminished (Figure 5c). Typically, small areas with higher temperature were visible for 30–80 s after release of the cuff. The appearance of the “hot-spots” depends on the geometry and size of local vascular units (perforosomes), the location of dominant perforator arteries (direct perforator), and their interconnects within perforosomes [11,12,13]. Interestingly, the site of thermal “hot-spots” partly corresponds to the location of light mottling patches observed during the last stage (10 min) of occlusion (see Figure 4), whereas dark bluish and purple areas were predominant in the “hot-spot” free regions. Such a tendency was observed in almost all subjects.

## 4. Discussion

The present study introduces a recently developed multimodal device prototype for dual-modality microcirculation assessment using visible range HSI and thermography. We have developed and tested the HSI module in the previous work [25], where we proved the reliability of the HSI system for in vitro and in vivo measurements in the 500–625 nm range. Here we upgraded the system, so that it can perform fast (real-time) analysis of multimodal data and can estimate a small number of microcirculation parameters. This was possible due to the built-in powerful computer with custom-made combinations of powerful Matlab image processing algorithms (image preprocessing, k-means clustering, analytical diffuse optical model with non-linear curve fitting, etc.). We tried to avoid complicated multi-layered skin models (e.g., Monte Carlo simulations) [23,24], and we showed that two-layered analytical skin model [30] can be reliable for skin microcirculation parameter estimation, using visible light for skin mottling and oxygen saturation assessment.

We have used a16 spectral channel snapshot HSI camera. The Ximea xiSpec camera sensor has Fabry–Perot filters, stretching the visible sensitivity due to spectral mixing [32]. We have taken into account that changing the numerical aperture of the lens mounted on the Ximea camera slightly changes the spectral response of the channels, which most likely will affect the precision of determining the blood oxygenation. Previously, we tested the spectral response of the HSI sensor using a monochromator setup, and we found that increasing the aperture leads to the widening of the spectral response of the sensor, and this leads to errors, which are negligible in the 507–625 nm range (unpublished data). We decided that aperture f2.8 is acceptable to get the maximum of light and the same time without affecting the precision of spectral responses of the camera sensor. The illuminator emission spectra were chosen to reach the maximum dynamic range of the HSI camera when the skin image is acquired. Therefore, we minimized spectral mixing, and at the same time, we focused on blood oxy/deoxy-hemoglobin absorption peaks.

Another problem is camera sensor noise, which leads to the unstable estimation of tissue parameters, and as a result, parametric maps show sparse noise [24,32]. Our system avoids this noise using the special clustering algorithm.

Both hyperspectral and thermal camera systems allow assessing skin microcirculation [33,34]. To validate the hypothesis, we performed measurements on healthy subjects during upper arm occlusion tests. The results confirmed the reliability of the developed system for in vivo assessment of skin blood oxygen saturation and temperature maps. Despite the fact that HSI systems are less sensitive to changes in oxygen saturation than to changes in blood content [23,24,32], our system was able to visualize changes of *SO*_2_ in a wide range (Table 2). Additionally, the results were comparable to other studies [24,34].

The tissue parameters measured by our system—oxygen saturation maps, mottling parameter, and temperature distribution—are fundamental descriptors of tissue that provide a snapshot of its physiology. The estimated parameters are suitable for tissue diagnostics in the clinical environment. The small-form factor, light weight, and low power consumption of our system allow the system to be carried by physicians who move between patients’ beds or between clinics. If successfully translated and productized, it can have a profound effect on clinical care.

It is still unknown how our system compares with more expensive spectral imaging systems in terms of performing scattering and absorption measurements. Also, there are some inherent challenges in working with a diffuse skin model, as the fitting analysis becomes an under-determined problem, where the estimation of scattering parameters is problematic. This could be solved if some parameters (e.g., melanin concentration) are set fixed, while others (blood volume and oxygen saturation) can be altered. This approach is feasible when the measurements are performed on the same subject during different physiological conditions (e.g., occlusion tests).

Dual-mode imaging system for non-contact imaging of cutaneous tissue oxygenation and vascular function could be very promising for appropriate detection and treatment of many health disorders such as chronic wounds, sepsis, or the monitoring of skin flaps during the reconstructive surgery. By using appropriate image processing algorithms, it is possible to fuse dynamic thermographic images with HSI maps, revealing a better view of skin circulation [34]. Nevertheless, further studies are needed in order to delineate a reliable quantitative correlation between the oxygen saturation and skin tissue temperature gradients.

In the future, we plan to improve our prototype device in terms of miniaturization and ease-of-use in the clinical environment. Many improvements will be done on algorithms and software. Afterwards, the upgraded device should be tested clinically. Additionally, the Monte Carlo look-up table approach [23,29] could be involved.

## Figures and Tables

**Figure 1 biosensors-09-00097-f001:**
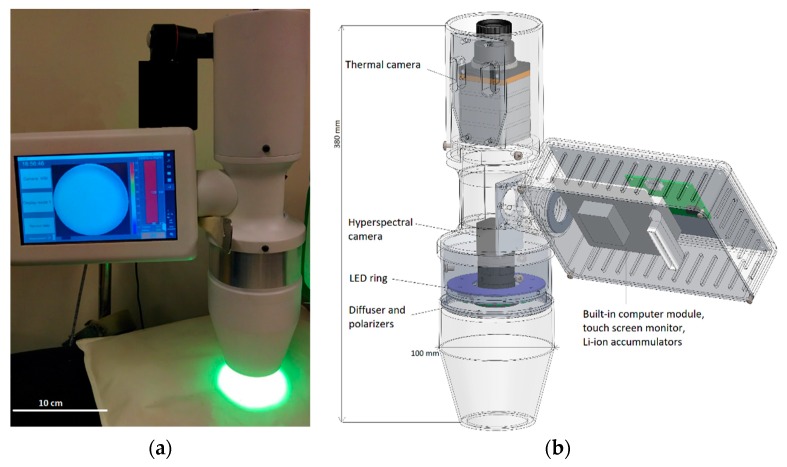
The hybrid prototype device (**a**), and its schematic image (**b**).

**Figure 2 biosensors-09-00097-f002:**
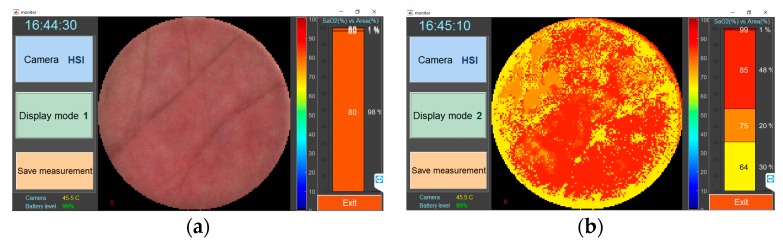
The screenshot of software: (**a**) Preview mode, and (**b**) oxygen saturation monitoring mode.

**Figure 3 biosensors-09-00097-f003:**
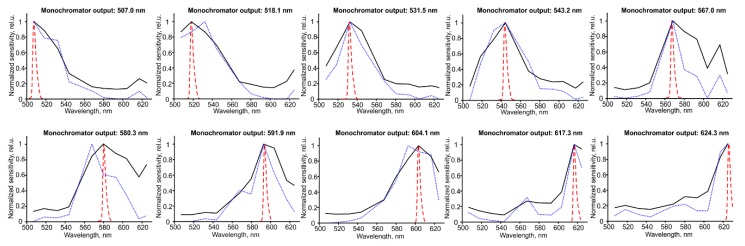
The responses of 5th to 14th spectral channels of HSI sensor (black line) to monochromatic illumination measured by spectrometer (dotted red line), and spectrally unmixed sensor responses, calculated from Ximea datasheet data (dotted blue line). Each subplot represents sensor response to the wavelength which corresponds to the nearest spectral channel.

**Figure 4 biosensors-09-00097-f004:**
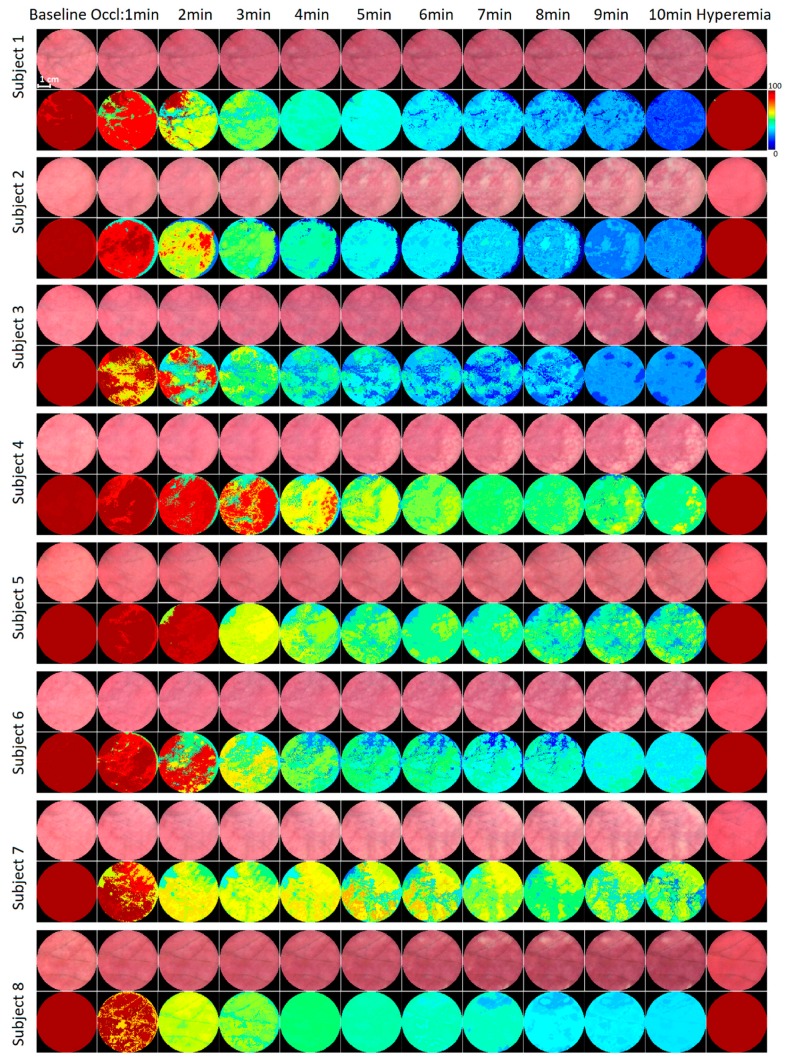
Skin images and clustered (*K* = 10) oxygen saturation maps during the standard occlusion tests of eight subjects.

**Figure 5 biosensors-09-00097-f005:**
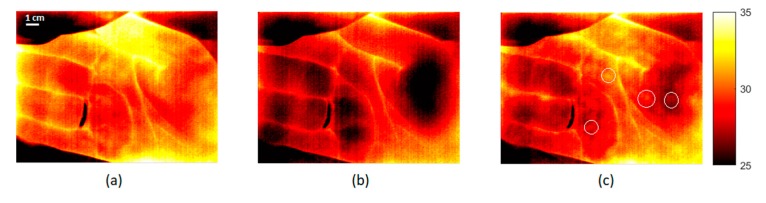
Thermal images of palm skin in baseline (**a**), after the 10-min long occlusion (**b**), and after the removal of cuff (**c**). White circles emphasize “hot-spots” during the hyperemia stage.

**Table 1 biosensors-09-00097-t001:** Model parameters [31].

Skin Layer	*B **	*SO*_2_*** [%]	*M **	*µ’_s_* _500 nm_	*f_Ray_*	*b_Mie_*	*A*	*d_e_* [µm]
Epidermis	-	-	10^−3^–10^3^	48	0.4	0.7	0.2	60–120
Dermis	10^−8^–10^−2^	0–100	-	48	0.4	0.7	0.2	∞

* Variable parameter.

**Table 2 biosensors-09-00097-t002:** Calculated oxygen saturation % values, during the provocation tests: Baseline (B), occlusion (1–10 min), hyperemia (H). Group-averaged data from 8 subjects.

	Baseline		Occlusion (minutes)								Hyperemia	
	B	1	2	3	4	5	6	7	8	9	10	H
*SO* _2 (mean)_	99.8 ± 0.7	90.1 ± 5.2	69.8 ± 14.4	55.8 ± 9.2	49.1 ± 8.1	45.0 ± 8.3	42.9 ±9.4	40.6 ± 9.6	38.5 ± 9.7	37.6 ± 10.7	36.1 ± 12.2	100 ± 0.0
*SO* _2 (Std)_	0.6 ± 0.7	13.9 ± 4.2	14.5 ± 7.0	9.4 ± 5.1	8.0 ± 4.3	8.0 ± 3.7	7.5 ± 3.5	7.6 ± 3.4	7.5 ± 2.5	6.5 ± 2.9	6.1 ± 2.9	0.1 ± 0.4

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
