# Peer review of "Multimodal Device for Real-Time Monitoring of Skin Oxygen Saturation and Microcirculation Function"

_biosensors, 2019, doi:10.3390/bios9030097_

Round 1

Reviewer 1 Report

The authors present a compact hybrid device combining hyperspectral and thermal imaging for functional characterization of human skin in vivo. The occlusion tests have been used to validate system performance. The paper adequately describes the development and testing of the device. However, it leave a space for the improvements and corrections.  

1. Please specify what is the irradiance level on the skin surface during the measurements?

2. Specify the field of view diameter and the spatial resolution of the system in both cases for hyperspectral and thermal imaging.

3. What are the data acquisition time and data processing time in both cases?

4. Please explain in more details what is the meaning of A matrix in the eq. 1.

5. Same for wi in the e.q 7.

6. The authors have reported that the measured blood oxygenation in the base line is close to 100%. This is valid for the arterial blood, however, the skin contains venous blood as well with the oxygenation of 80-70%. I suppose that you measure the averaged value of the oxygenation of the mixed blood. Did you apply any calibration/correction?

7. Correct the table 2 so that the values are placed within one line. Currently, it is quite hard to read.

8. What are the star-like or branch-like structures in fig 4? Unfortunately, I cannot distinguish them.

9. On page 9, the authors report about the scattering parameter that has been measured. What it actually is?

10. Add scale bars to fig 3 and fig 4.

11. On page 9, the authors discuss about the comparison of the developed system to more expensive spectral imaging systems. Could you please estimate the cost of your system? Please provide also the links to the systems you are talking about.

12. Please perform a professional language check as the current manuscript contains some stylistic issues, grammar mistakes and typos.

Reviewer 2 Report

My main concerns with the presented results are:

There is a the lack of validation of the method for measuring superficial skin blood oxygen saturation. Previous experiments referred to (ref # 25) were sampling digit (finger) capillary blood. owever, it is not clear whether this blood is really representative of finger skin superficial blood. There are a couple of suggested experiments that have been described in the literature: the authors could mix intralipid (bulk scattering) and blood to make a homogenous tissue phantom where scattering level can be the in same range in the actual spectral range for the detector, as that of human skin. By oxygenizing and deoxygenizing the blood by e.g. bubbling pure nitrogen in the mixture. Reference saturation can be measured with pO2 electrode. This type of experiment is needed Before values can be trusted.   

The XIMEA sensor handling is not described in detail. At what aperture stop was the sensor used? The spectral beavior of a Fabry-Perot filter depends on incident angle to the filter, which in turn depends on the aperture. The spectra demixing suggested by the factory depends on these settings. Furthermore, was the spectral beavior of the filters checked with e.g. a monochromator setup?

A SO2 of 99.8% at baseline for skin is simply not correct, but rather matches that of arterial blood.   

Reviewer 3 Report

The paper “Multimodal device for real-time monitoring of skin saturation and microcirculation function” by U. Rubins et. al. introduced their recently developed compact hybrid device for real-time monitoring of skin oxygen saturation and temperature distribution. Further upper arm occlusion tests on 8 healthy volunteers were conducted, confirming the reliability of developed system for in vivo assessment of skin oxygen saturation and microcirculation.

The paper is very well written, with all the details needed for readers to know well their techniques and applications. I would suggest acceptance of the paper in its current form for publication in Biosensors.

Author Response

Dear reviewer. Thank You very much for the suggestion of acceptance of the paper.

Reviewer 4 Report

The manuscript presents a newly developed portable device for real-time hyperspectral and thermal imaging of the skin. The authors performed a validation study in vivo on the palm skin by introducing vascular occlusion. The manuscript is overall well written. Below I have some minor comments that the authors are suggested to consider.

1. The “skin saturation” in the title could be confusing. The authors are suggested to make it more specific, e.g. skin oxygen saturation.

2. The vascular occlusion site is not clear. In some parts of the manuscript, the descriptions show “upper arm”, which usually mean the portion between the shoulder joint and elbow joint, but in other parts of the manuscript, the authors claim “forearm” that refers to the portion between wrist and elbow joint. Please clarify this.

3. In Figure 1, for either the picture or the schematic image, the authors are suggested to provide a scale so that the readers can have a direct understanding of the size of this device.

4. The authors state “For each cluster the SO2 is calculated (equation 3)”. Should it be equation 4?

5. At the end of Results section, the authors connect the thermal image with the oxygen saturation image. This is an interesting observation, but it is not very easy to see what the authors describe. I think it would be helpful if the authors could use annotations (e.g. arrows) in the figures to more clearly point out the connections.

6. It could be helpful if the authors can add several sentences to comment or discuss the possibility and potential benefits (if any) of performing simultaneous or colocalized hyperspectral and thermal imaging?

Round 2

Reviewer 2 Report

The validation experiment I refer to is not asking for trancutaneous pO2, simply pO2 by such an electrode for use in a liquid solution. This independent measurement is needed before you can say that the method is validated. The validations refered to in paper #8 (numbering in reply to my comment),) uses a different camera (Nuance EX) with a very different technique (liquid crystal tunable filter I presume). A LCTF filter gives a very different, much cleaner spectral response, than that of a filter mosaic. Hence , the present camera is not validated. The authors could also easily repeat a finger occlusion of that experiment or choos to do brachial occlusion and study forearm skin (or palmar skin).

A better presentation of the results of the monochromator setup results is needed beofre we van appreciate that the differece from manufacturer spec is small.

Author Response

Dear reviewer.

Please see the attachment with response to your comments.
